# Impact of Early Nutrient Intake and First Year Growth on Neurodevelopment of Very Low Birth Weight Newborns

**DOI:** 10.3390/nu14183682

**Published:** 2022-09-06

**Authors:** Rasa Brinkis, Kerstin Albertsson-Wikland, Rasa Tamelienė, Ilona Aldakauskienė, Inesa Rimdeikienė, Vitalija Marmienė, Kastytis Šmigelskas, Rasa Verkauskienė

**Affiliations:** 1Department of Neonatology, Lithuanian University of Health Sciences, 44307 Kaunas, Lithuania; 2Department of Physiology/Endocrinology, Institute of Neuroscience and Physiology, Sahlgrenska Academy, University of Gothenburg, 40530 Gothenburg, Sweden; 3Department of Rehabilitation, Lithuanian University of Health Sciences, 44307 Kaunas, Lithuania; 4Department of Psychiatry, Hospital of Lithuanian University of Health Sciences, 50161 Kaunas, Lithuania; 5Health Research Institute, Faculty of Public Health, Lithuanian University of Health Sciences, 44307 Kaunas, Lithuania; 6Institute of Endocrinology, Lithuanian University of Health Sciences, 44307 Kaunas, Lithuania

**Keywords:** very low birth weight, newborn, enteral feeding, nutrient intake, growth, neurodevelopment

## Abstract

Optimal nutrient intake ensuring better neurodevelopment for very low birth weight (VLBW) infants remains unknown. The aim of this study was to assess the relationship between early (first 28 days) nutritional intake, first year growth, and neurodevelopment. In total, 120 VLBW infants were included into the study. A group of 95 infants completed follow-up to 12 months of corrected gestational age (CGA). Nutrient intake was assessed, and weight, length, and head circumference (HC) were measured weekly until discharge and at 3, 6, 9, and 12 months of CGA. Neurodevelopment was assessed at 12 months of CGA. Two groups—extremely preterm (EP) and very/moderately preterm (VP)—were compared. Growth before discharge was slower in the EP group than the VP group. At 12 months, there was no difference in anthropometric characteristics or neurodevelopmental scores between the groups. Higher carbohydrate intake during the first 28 days was the single significant predictor for better cognitive scores only in the EP group (β_s_ = 0.60, *p* = 0.017). Other nutrients and growth before discharge were not significant for cognitive and motor scores in either group in multivariable models, whereas post-discharge HC growth was associated with both cognitive and motor scores in the VP group. Monitoring intake of all nutrients and both pre-discharge and post-discharge growth is essential for gaining knowledge about individualized nutrition for optimal neurodevelopment.

## 1. Introduction

Survival of very low birth weight (VLBW) infants has increased over the last few decades; however, poor postnatal growth and suboptimal neurodevelopmental outcomes are common in this population compared to the infants born at term [1,2,3]. Preterm birth alters brain development and multiple factors have been shown to have an impact on neurological outcomes: preexisting conditions, size at birth [4], gestational age (GA) [5], nutrition and postnatal growth faltering [6], and comorbidities [7,8,9,10]. Preterm infants are born during a period of rapid fetal growth, requiring high nutrient intake to promote extrauterine growth. For many years, it has been widely accepted that preterm infants should mimic the growth of the fetus in utero. Nevertheless, many preterm infants fail to follow this pattern [11]. Observational growth studies found a positive association between postnatal growth and later neurodevelopment, while randomized clinical trials reported limited or no effect [12]. Defining postnatal growth failure in prematurely born infants—both before- and after-term—depends not only on the reference used but also on the interpretation of different estimates [13,14,15], which may subsequently lead to inappropriate nutritional interventions.

Providing recommended high nutrient intake is challenging in clinical settings. Parenteral nutrition (PN) may prevent nutritional deficits during critical periods of life, and numerous studies have assessed the effects of parenteral nutrient intake, especially protein, on growth and neurodevelopment [16,17,18]. Lack of evidence that higher amino acid intake may improve short and long term outcomes has led to a revision of parenteral nutrition recommendations [19,20], lowering previously recommended amino acid intake for preterm, and nutrient and energy intake for critically ill infants. Recent data show that prolonged parenteral nutrition and higher parenteral nutrient and energy intake may be associated with worse neurodevelopment [21,22]. More evidence suggests that early enteral nutrient and energy intake is associated with better cerebral growth, head growth, brain structure, and neurodevelopment [21,23,24]. Moreover, early enteral feeding may improve tolerance of PN [25], reducing adverse metabolic effects, such as hyperglycemia, which also may be related to neurodevelopmental impairment [26]. Thus, current nutritional strategies recommend early enteral feeding with human milk (HM) and enhanced nutrient provision with HM fortification [27,28]. Results from randomized studies comparing higher versus lower enteral protein intakes report inconclusive effects on early growth, and only a few studies reported on neurocognitive outcomes [29,30]. Enteral macronutrient proportions are more difficult to adjust than parenteral, and there is a lack of data regarding optimal fortification [31,32,33].

The aim of this study was to assess the relationship between nutritional intake during the first 28 days and weight, length, and head circumference (HC) growth during the first year of life and neurodevelopment at 12 months of corrected gestational age (CGA) of VLBW infants receiving early progressive enteral feeding.

## 2. Materials and Methods

### 2.1. Study Subjects

Ethical approval: The study was conducted at the Hospital of Lithuanian University of Health Sciences, Department of Neonatology. Approval for the study was obtained at the Kaunas Regional Bioethics Committee (approval no. BE-2-12). The study was registered at ISRCTN Database (no. ISRCTN64647571). Inclusion criteria for study participants were birth weight < 1500 g and GA ≤ 34 weeks, and written consent of both parents obtained. Exclusion criteria were chromosomal abnormalities, genetic syndromes which may affect growth, absent parental consent, and surgical intervention with partial bowel removal.

In total, 120 VLBW infants (63 girls and 57 boys) born between 31 May 2018 and 17 May 2020 were included into the study.

To identify which nutritional and growth data might play a role for later neurodevelopment, we analyzed growth patterns of weight, length, and HC in each gestational week and compared two groups of infants—extremely preterm (EP), born 23–27 weeks of gestation; and very/moderately preterm (VP), born 28–34 weeks of gestation.

### 2.2. Nutritional Practices

Infants were started on both parenteral (PN) and enteral nutrition (EN) right after birth. For PN 2.5–3.0 g/kg of amino acids, 6.0–7.0 g/kg carbohydrates, 1.0 g/kg lipids, 45–50 kcal/kg and 70–80 mL/kg of fluids were provided on the first day of life, using standard mixture. If the EN could not be advanced or interrupted, PN was increased up to target intake 3.5–3.8 g/kg/day of amino acids, 12.0–16.0 g/kg/day of carbohydrates, 3.0 g/kg/day of lipids, and 80–90 kcal/kg and 140 mL/kg of intravenous fluids [19].

Enteral feeding with 20 mL/kg/day of own mother’s milk (OMM) or donor milk was started at a median time of 3–6 h after birth and advanced rapidly by 20–30 mL/kg day [34]. Standard method of HM fortification was started when full enteral nutrition was reached, aiming to provide recommended enteral nutrient intake [35], and was used throughout hospitalization period. With advancing milk intake, PN weaning began at day 3 to 4, maintaining gradual increase in total daily fluid intake at 20–25 mL/kg/day up to 150–160 mL/kg/day at the end of the first week [36]. Median time to reach full enteral feeding was 7 days. By day 14 of life, all infants were on full enteral feeding. Human milk analysis was performed twice weekly for macronutrient content and daily total nutrient intake was calculated for the first 28 days. Detailed nutritional practices applied to the study cohort and nutrient intake calculations were described previously [34]. Average daily total protein, carbohydrates, and fat intake for the first 28 days of life were used for this study analysis. Few data were available on the first week’s mother’s milk composition because of rapidly advancing enteral feeding with fresh OMM. Mothers were able to provide extra milk for analysis only by the end of the first week. Therefore, total daily enteral nutrient intake during the first week of life is underestimated, leading to slightly underestimated mean daily intake for the whole period of 28 days. Of the infants, 92% were fed only OMM at discharge. After discharge, infants were fed either OMM or standard term infant formula and no post-discharge fortification was administered.

### 2.3. Anthropometric Measurements

Weight, length, and HC measurements were performed weekly from birth to discharge and at 3, 6, 9, and 12 months of CGA following the standard procedure [37]. Infants were weighed using incubator scales (Giraffe, GE Healthcare, Laurel, MD, USA) while in the incubator care, and portable electronic scales (Marsden, Rotherham, UK) later. Body length and head circumference were measured to the nearest 0.1 cm. An infant measuring rod (SECA, Hamburg, Germany) was used for length measurement, and non-stretchable single-use tape (SECA, Hamburg, Germany) was used for HC measurement. Average value of two length and HC measurements were used for analyses. Measurements were performed by two investigators to reduce errors. Median time at discharge was 36 weeks of CGA; thus, discharge anthropometric measurements were used of completed gestational weeks if discharge occurred before 36 weeks of CGA and at 36 weeks if discharge occurred at that time or later. Weight, length, and HC were calculated in SDS, using the only continuous growth reference for birth size and infancy growth, which is based on a healthy population, one for birth and one for postnatal growth [14]. SDS in this reference is equal to a z-score used in other references, both adjusted for age and sex.

### 2.4. Neurodevelopment

Neurodevelopment assessment was performed at 12 months of CGA by trained specialists using Bayley Scales of Infant Development, Second Edition (BSID-II) [38]. Each infant was assessed once. Cognitive (Mental Development Index (MDI)) and motor (Psychomotor Development Index (PDI)) scores were classified as normal if they were within 1 standard deviation (SD) from mean (mean = 100, SD = 15), i.e., ≥85. Mild development delay was diagnosed when the scores were between −1 SD and −2 SD (≥70 and <85); moderate delay, scores between −2 SD and −3 SD (≥55 and <70); and severe, more than 3 SD below the mean (<55). When the infant failed to achieve the minimum score of 50, a value of 49 was given.

### 2.5. Statistical Analysis

Data analysis was performed using Microsoft Excel version 16.54 and IBM SPSS Statistics for Windows (version 27.0. IBM Corp., Armonk, NY, USA). The descriptive analysis included means and standard deviations (±SD), medians with interquartile ranges (IQR), and numbers and percentages for categorical indicators. For comparisons of normally distributed values *t*-test, for non-normally distributed values—non-parametric Mann–Whitney test and for associations of categorical indicators, the chi-squared test were used.

To predict neurodevelopment, multiple linear regression models were created. The strength of the factors in the model was assessed using crude (B) and standardized beta (β_s_) coefficients. The model fit was estimated using R^2^ coefficient. The results were considered statistically significant when *p*-value was < 0.05.

## 3. Results

### 3.1. Sample Characteristics

In total, 95 infants returned to follow-up at 12 months CGA, constituting 79% of the initial cohort and 87% of survivors without bowel resection. For non-survivors (*n* = 8) and infants with bowel resection (*n* = 3), median gestational age and birth weight were 25 weeks and 700 g. The follow-up flowchart of study participants is shown in Figure 1.

There were no differences in the main demographic characteristics between infants who did and who did not return to follow-up. The exception was that more girls than boys did not return to the follow up. Main demographic characteristics of the follow-up infants are shown in the Table 1.

### 3.2. First Year Growth and Neurodevelopment

EP infants had a decline in SDS of all growth parameters for approximately 4 weeks after birth, while VP regained growth trajectory after the first week of life. Growth patterns of both groups are shown in Figure 2.

Weight and HC growth during the hospitalization period had different patterns in EP and VP infants, especially during the first four weeks, while length growth did not differ. EP infants started to accelerate in all growth parameters from the fourth week onwards, and by the time of discharge there were no differences in weight SDS, length SDS, and HC SDS between the two groups. At discharge, EP infants’ HC reached birth size SDS, and length exceeded birth length SDS, only weight SDS remained below birth value. In the VP group, weight and HC SDS reached birth values, and length exceeded birth length SDS. After discharge, both groups showed accelerated weight and HC growth to 3 months of CGA, and length growth until 12 months of CGA. At 12 months of CGA, there were no differences in weight SDS, length SDS, and HC SDS between the groups. At 12 months of CGA, both groups were at zero SDS for length, −1 SDS for weight, and −0.5 SDS for HC.

To compare the findings based on SDS values with other references, the growth curves were also calculated using z-score values by Fenton [15] during hospitalization and using WHO reference [39] from 3 to 12 months. The growth patterns based on these references are in the Appendix A.

Nutritional intake was not different between the groups during first 28 days. Extremely preterm infants had longer duration of parenteral support (*p* < 0.001), higher parenteral nutrient intake during the first week of life. Slower advancing of enteral feeding led to later start of fortification in EP group, and lower intake of protein and carbohydrate during the second week. However, these differences did not influence average total nutrient and energy intake during the whole first 28 days period. Higher recommended protein intake for EP was not achieved with standard fortification. More detailed nutritional calculations are presented in the Appendix A.

Neurodevelopment, as MDI and PDI scores, did not differ at 12 months of CGA between the groups. Growth, nutritional, and neurodevelopment data are shown in Table 2 and Table 3.

At 12 months of CGA, 52.5% of follow-up infants in the EP group and 47.1% in the VP group had both normal cognitive scores and normal motor scores, 10% and 11.8% infants had mild delay of both cognitive and motor scores. In both groups, there were 25% of infants who had normal cognition but mild motor delay. One infant (1.1%) in the EP group had severe cognitive and motor delay and was diagnosed with cerebral palsy. No infants had unilateral or bilateral blindness and no infants were diagnosed with hearing impairment requiring intervention.

### 3.3. Relationship between Nutrition, Growth, and Neurodevelopment

To estimate which nutritional and growth indices may predict later neurodevelopment, two multiple linear regression models were created. Model 1 included characteristics prior to discharge, i.e., the model was built to predict neurodevelopment at 12 months of CGA. In contrast, Model 2 included characteristics up to 12 months, therefore, the model was built to explain which growth parameters had potential impact on neurodevelopment at 12 months of CGA.

In Model 1, sex, GA, and size at birth (birth weight SDS), growth during hospitalization, and mean daily nutrient intake were included. Higher intake of carbohydrates was the only significant predictor of better mental development in extremely preterm infants (β_s_ = 0.60, *p* = 0.017) unlike in very/moderately preterm infants, for whom none of the nutrients were associated with MDI. Change in SDS for weight, length, and HC from birth to discharge were not associated with MDI and PDI scores in any of the groups (Table 4).

The same model was made using growth during hospitalization expressed by z-scores using Fenton reference [15] and it is presented in Appendix A. In the model based on Fenton growth reference, carbohydrates intake also was the only significant predictor of better mental development in extremely preterm infants (β_s_ = 0.62, *p* = 0.008). Change in z-scores for weight, length, and HC from birth to discharge were not associated with MDI and PDI scores in any of the groups.

In Model 2, where post-discharge growth and early nutrient intakes were included, carbohydrates remained the strongest predictor of MDI in EP group, although, unlike in Model 1, it was at the borderline of significance (β_s_ = 0.49, *p* = 0.053). Among post-discharge growth parameters better head circumference growth was associated with both better MDI (β_s_ = 0.37, *p* = 0.056) and better PDI (β_s_ = 0.39, *p* = 0.053) at the borderline of significance, and only in very/moderately preterm infants (Table 5).

## 4. Discussion

Neonatal care results in an increasing number of less mature preterm infants surviving into adult life. However, it remains unknown what nutrient intake ensures normalized postnatal growth trajectories and neurodevelopment for very low birth weight (VLBW) infants. To gain knowledge and evaluate whether recently proposed early progressive enteral feeding strategy will result in improved growth and neurodevelopment as a long-term goal, we undertook the present single center study with preterm infants born in the period of two years. Nutrient intake was assessed during the first 28 days of life for its impact on first year growth and neurodevelopment. Besides parenteral nutrition, the infants in our study cohort were started to feed enterally early, within the first hours after birth, mimicking a healthy infant’s feeding. Even with morbidities, they tolerated feeding well. Recommended enteral nutrient intake [35] was reached on the second week of life, which ensured catchup growth for weight, length, and HC during first year of life up to normalized postnatal growth patterns.

The main finding of this study is that higher carbohydrate intake, but not protein, which is being explored most extensively as key nutrient in preterm population, was the only significant nutritional predictor of better cognitive scores, but only in extremely preterm infants. Another important finding is that despite different early postnatal growth patterns between these groups, growth during infancy and neurodevelopment outcomes at 12 months of CGA were similar in both groups. At 12 months of CGA, extremely and very preterm infants had similar proportions of suboptimal neurodevelopment, despite very preterm infants growing faster before discharge and having similar nutritional intake.

### 4.1. First Year Growth and Neurodevelopment

The major differences in growth of EP and VP newborns are seen early after birth, during the first four weeks of life, between EP and VP groups. Changes in weight, length, and HC SDS were positive during hospitalization and post-discharge periods in the very/moderately preterm group. The EP group showed a different growth pattern from VP and grew slower during the first four weeks (when most morbidities occur). However, their growth accelerated between 4 weeks and discharge, and thereafter, resulting in no difference in weight, length, and HC between the groups at 12 months of CGA, both when expressed in SDS and in attained weight, length, and HC. Postnatal growth assessment remains challenging due to variety of the references used and need to switch between the references after term age. Most infants in our cohort were discharged before term, and growth data at 40 weeks of CGA were lacking. To be consistent with growth assessment throughout the follow-up period, we used the continuous growth reference for size at birth and postnatally during infancy. This is based on 800,000 healthy Swedish infants out of the about 1 million born in 1990–1999. The populations in Lithuania and Sweden nearly ended the secular trend, resulting in similar birth size and adult height. It is important to note, that using a “healthy populations” reference—omitting infants born as twins, with syndromes, diseases, or as stillbirth [14]—for birth size, SDS for all growth parameters at birth became lower compared to using Fenton reference since it is based on nearly the whole population [15]. The other reasons for preterm infants to have lower SDS are that infants born preterm might have undergone some extent of intrauterine growth restriction related to pregnancy complications resulting in preterm birth [11], and in our cohort VP infants had lower birth weight SDS than EP. Using different growth references will provide different values for birth size and growth evaluation. When the goal is to compare different studies, the same reference, regardless which one, shall be used. For this purpose, we also provided growth patterns based on Fenton [15] and WHO [39,40] references, since recent neonatal nutrition assessment indicators are based on Fenton reference [41]. However, our main purpose was to compare the growth of our preterm infants with the growth of population as healthy as possible. Our cohort’s growth trajectories, estimated from a reference of “healthy population”, show that the VP group regained the birth weight SDS and exceeded it by the time of discharge, and exceeded birth SDS of all three growth parameters at 12 months of CGA. The rapid weight catch-up growth carries its own risks however [42]. In both groups, infants demonstrated length and HC acceleration as well. Changes in all three individual growth parameters should be evaluated looking for optimal growth pattern as a marker of good neurodevelopment and considering potential risks of later metabolic consequences. Observational studies reported positive association between postnatal weight gain, HC growth, and later neurodevelopment, while randomized clinical trials reported limited or no effects [12]. Our regression models revealed that weight gain, length, and HC growth before discharge were not associated with later neurodevelopment outcomes, independently of any of the two growth references used, contrary to the results of study by Ehrenkranz [6] reporting that infants with faster weight gain during Neonatal Intensive Care stay had lower incidence of cerebral palsy and neurodevelopment impairment. One of the explanations of these differences might be that our cohort grew relatively well already during hospitalization, and it may be hypothesized that a certain threshold for growth velocity exists, above which later neurodevelopment is not affected. Better post-discharge head growth was associated with improved cognitive and motor scores, but only in the very preterm infants’ group. Despite better weight gain and linear growth, very preterm infants did not show better neurodevelopment scores than the extremely preterm infants.

### 4.2. Early Nutrient Intake and Neurodevelopment

A developing brain is vulnerable to undernutrition, thus, in recent years many studies have focused on improving nutritional practices, both enteral and parenteral [43]. Nutrient enriched early diets have been shown to improve short-term growth and neurodevelopment [44], but more recent data show that higher early protein and energy intake does not improve neurodevelopmental outcomes at 7 years [45]. Parenteral nutrition related adverse effects, such as hyperglycemia, are shown to be associated with poorer neurodevelopmental outcomes at 24 and 12 months of CGA [26,46]. More evidence is emerging regarding the advantages of enteral nutrients over parenteral [21] and well-known benefits of enteral feeding and breastmilk on cerebral growth, brain structures, and neurodevelopment [23,47]. Moreover, the study by Boscarino et al. found that early enteral feeding may improve tolerance of parenteral nutrition in preterm newborns and may prevent PN related metabolic complications, such as hyperglycemia, hypertriglyceridemia, and metabolic acidosis [25]. Early feeding with own mother’s milk is now a preferred practice for very preterm infants. In our study, infants were fed with predominantly fresh, unprocessed OMM from the first day of life. Parenteral nutrition was limited to the first week of life for most infants (61% and 93% in EP and VP infants, respectively), thus, enteral nutrient intake was dominant during the first 28 days. In our study, higher carbohydrate intake—not protein or fat intake—was the strongest predictor of better cognitive scores, but only in extremely preterm infants. None of the nutrients had any impact on neurodevelopment of infants born at 28–34 weeks of gestation. By calculating macronutrient intakes, we found that protein intake was falling below recommended levels from the week 4 and carbohydrate intake exceeded recommended values by the approximate levels coming from the fortifier [34]. In the early hospitalization period, both protein and carbohydrates were positively associated with weight gain and HC growth during the first 28 days, and although this correlation was weak in extremely low birth weight infants, we hypothesize that carbohydrates may have had an impact on HC growth acceleration after 4 weeks of life. HC growth is associated with brain growth. Glucose being the primary energy source for the brain possibly explains the underlying the importance of carbohydrates on MDI scores of extremely preterm infants. Many studies have focused mainly on protein and energy without specifying non-protein energy source intakes, and adjustment of enteral nutrient intake can only be made with amount of milk or fortifier, which makes adjusting nutrient proportions even more challenging in the daily clinical practice. The composition of human milk is dynamic [48], and different HM fortifiers have different macronutrient composition. Non-protein energy comes from both carbohydrates and fat, and we considered it important to analyze each nutrient separately. We found only one study assessing relationship between carbohydrates intake and growth conducted by Collins et al. [49]. They found that carbohydrates intake was the main determinant of growth of very preterm infants when protein intake was sufficient; however, in this study, reference values for human milk composition were used and neurodevelopment was not assessed. In our study protein intake did not have any relationship with either MDI or PDI scores and its intake was lower than recommended for extremely low birth weight infants, using standard fortification. Results from other studies focusing on protein intake are inconclusive—some authors report positive effect of protein on neurodevelopment [30,44], others report the opposite findings [50]. The latter study found that higher total protein intake resulted in lower motor scores at 2 years of CGA in ELBW infants, despite improved white matter integrity; however, the duration of parenteral nutrition in this study was longer than ours and parenteral protein may have played a role.

We did not include energy into the model for collinearity reasons. However, studies analyzing energy intake found that the first week enteral energy intake is positively associated with cerebral growth, while parenteral energy results in poorer cerebral growth [23]. A study by Schneider et al. found that early enteral intake, lipids, and energy were associated with improved brain growth and PDI, but not MDI at 18 months [51]. Fat intake had no relationship with neurodevelopment in our cohort. Enteral fat intake in our study came solely from human milk and the role of lipids on growth and neurodevelopment development remains unclear.

Our feeding practices with early colostrum and fresh unprocessed own mothers’ milk mimicking healthy infant feeding may be associated with better cognition scores as demonstrated in other studies with term and preterm infants [52,53]. This suggests that the quality of nutrients may outweigh the quantity and it raises another concern regarding fortification strategies. We hypothesize that, like certain growth velocity thresholds, there may be a threshold for nutrients, above which neurodevelopment is not further improved, especially when early nutritional deficits are eliminated.

Our study has strengths and limitations. The strength of the study is nutritional calculations of all macronutrients, based on twice weekly human milk analysis and no reference values used. This makes our calculations and analysis more precise and reliable. To our knowledge, this is the first study assessing early carbohydrates’ intake on neurodevelopment of VLBW infants. Another strength is the analysis of not only growth measurements during hospitalization, but also post-term, during infancy. However, we did not obtain data at 40 weeks of CGA, since most infants were discharged earlier. Another limitation is also not monitoring post-discharge nutrition. A small and heterogenous sample did not allow us to explore the effects of parenteral and enteral intakes separately, as well as other factors which may influence later neurodevelopment, such as morbidities and intrauterine growth restriction. Assessment of neurodevelopment by Bayley II scales restricts comparison with many other recent studies, using Bayley III. Bayley II has been shown to result in lower scores, especially MDI, and be less concordant with other scales, such as Griffiths [54]. Finally, our cohort was assessed at early age—i.e., 12 months CGA instead of 24 months—which might also affect the conclusions, but further follow-up of this cohort is still ongoing.

## 5. Conclusions

In this study, we found dietary carbohydrates intake to be strong independent predictor of mental development in extremely preterm infants fed with early progressive enteral feeding. Further research with larger cohorts will be needed to investigate the effects of recently suggested nutritional interventions, such as early enteral feeding with human milk, timing, and necessity of fortification. The results of our study indicate the importance of analyzing all macronutrients provided by both parenteral and enteral routes, as well as all nutrients coming from fortifiers. Only careful monitoring of the effects of different nutrients on short- and long-term outcomes may allow us to optimize feeding practices of this population, especially since we are aiming to individualized feeding. Here, we used infancy growth as an explanatory variable for early neurodevelopment and found that growth during the hospitalization period might be less predictive on neurodevelopment than post-discharge growth. Thus, close post-discharge growth monitoring during infancy and onwards will be essential for gaining more knowledge of optimal nutrient intake in preterm infants during extrauterine life.

## Figures and Tables

**Figure 1 nutrients-14-03682-f001:**
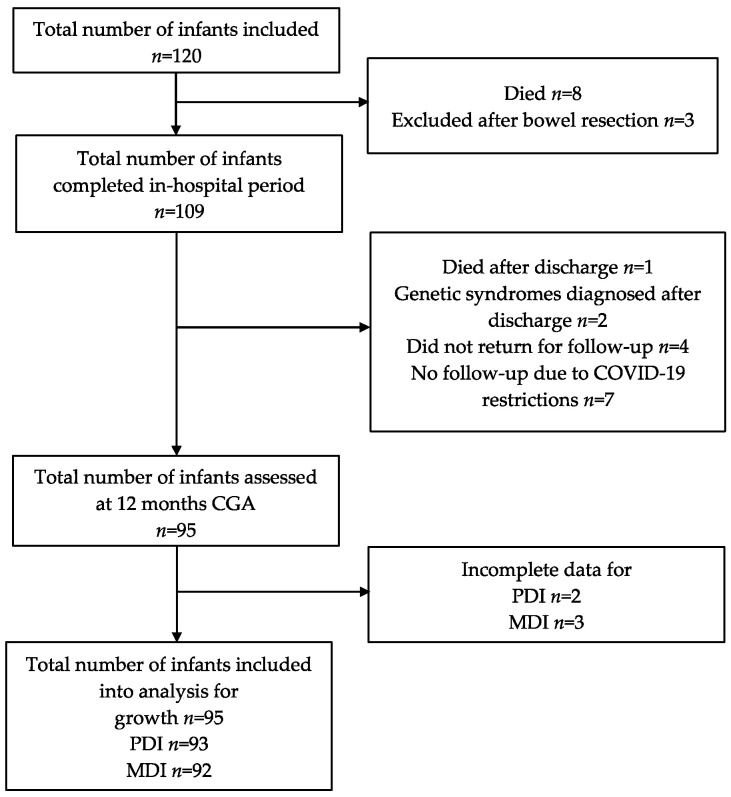
Follow-up chart of study participants.

**Figure 2 nutrients-14-03682-f002:**
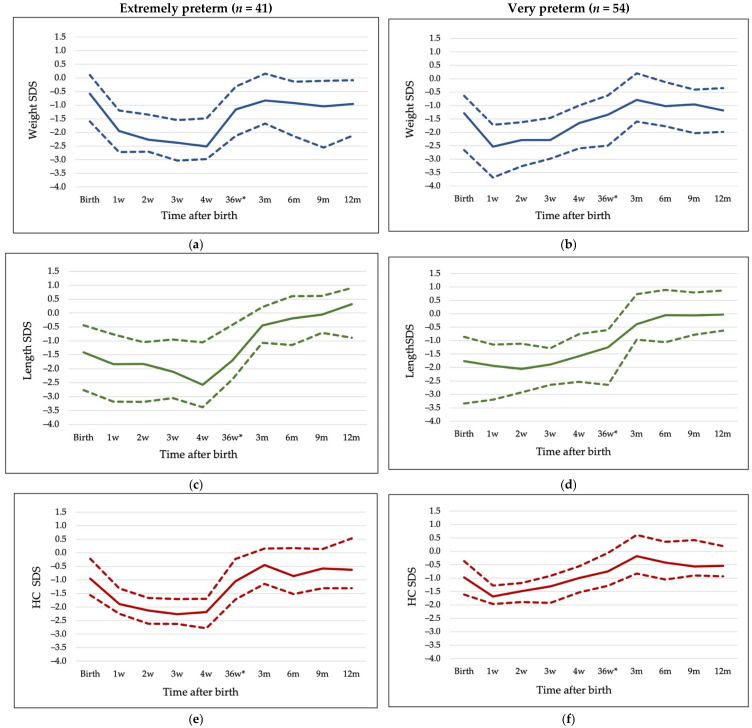
Growth patterns of groups during the first year of life. (**a**) Weight SDS in EP group, (**b**) Weight SDS in VP group, (**c**) Length SDS in EP group, (**d**) Length SDS in VP group, (**e**) Head circumference SDS in EP group, (**f**) Head circumference SDS in VP group. Values are median (solid line) and interquartile range (IQR) (dashed lines). 36w*—36 weeks of CGA or discharge.

**Table 1 nutrients-14-03682-t001:** Main demographic characteristics of the follow-up infants.

Characteristic		Returned to Follow-Up (*n* = 95)	No Follow-Up (*n* = 14)	*p*
Birth weight, g		1134 (930–1300)	1194 (1022–1322)	0.562
Gestational age, weeks		28 (26–29)	28.5 (27.5–30)	0.39
Male, *n* (%)		48 (50.5)	3 (21)	0.042
Female, *n* (%)		47 (49.5)	11 (79)	
Apgar score	1 min.	7 (5–8)	8 (5–8)	0.672
	5 min.	8 (7–9)	8 (8–9)	0.690
IVH ≥ III, *n* (%)		4 (4.2)	1 (7.1)	0.624
Sepsis, *n* (%) *		33 (34.7)	2 (14.3)	0.126
NEC, *n* (%)		3 (3.2)	0 (0)	N/A
BPD, *n* (%)		7 (7.4)	0 (0)	N/A

* Sepsis—early and late onset. IVH—intraventricular hemorrhage, NEC—necrotizing enterocolitis, BPD—bronchopulmonary dysplasia. Values are median (interquartile range) and *n* (%).

**Table 2 nutrients-14-03682-t002:** Characteristics of growth, nutrition, and neurodevelopment.

Characteristics	Extremely Preterm (*n* = 41)	Very/Moderately Preterm (*n* = 53)	*p*
Growth before discharge			
Weight SDS at birth	−0.58 (−1.59–0.11)	−1.29 (−2.66–−0.64)	0.002
Length SDS at birth	−1.42 (−2.77–−0.44)	−1.77 (−3.34–−0.86)	0.109
HC SDS at birth	−0.95 (−1.57–−0.22)	−0.97 (−1.61–−0.36)	0.499
Change in weight SDS birth to discharge	−0.44 (−1.08–0.30)	0.01 (−0.48–0.61)	0.007
Change in length SDS birth to discharge	−0.06 (−0.85–0.84)	0.32 (−0.42–0.96)	0.151
Change in HC SDS birth to discharge	−0.06 (−0.05–0.43)	0.38 (−0.29–0.89)	0.021
Weight SDS at discharge	−1.16 (−2.12–−0.31)	−1.34 (−2.50–−0.62)	0.199
Length SDS at discharge	−1.70 (−2.38–−0.42)	−1.25 (−2.64–−0.61)	0.775
HC SDS at discharge	−1.05 (−1.72–−0.23)	−0.75 (−1.28–−0.06)	0.151
**Growth after discharge**			
Change in weight SDS discharge to 12 months CGA	−0.05 (−0.86–0.71)	0.45 (−0.28–1.35)	0.07
Change in length SDS discharge to 12 months CGA	1.36 (0.78–2.17)	1.78 (0.51–2.64)	0.462
Change in HC SDS discharge to 12 months CGA	0.25 (−0.23–1.28)	0.18 (−0.34–0.89)	0.435
Weight SDS at 12 months CGA	−0.96 (−2.12–−0.08)	−1.18 (−1.98–−0.35)	0.645
Length SDS at 12 months CGA	0.32 (−0.89–0.90)	−0.03 (−0.63–0.87)	0.945
HC SDS at 12 months CGA	−0.63 (−1.31–0.53)	−0.54 (−0.94–0.19)	0.623
Weight at 12 months CGA, g Boys Girls	9500 (8630–10,400) 8965 (7690–9883)	9480 (8415–10,110) 8700 (8100–9595)	0.536 0.784
Length at 12 months CGA, cm Boys Girls	77.8 (74.0–79.0) 75.0 (72.4–77.0)	77.0 (75–78.8) 74.0 (73.0–77.0)	0.992 0.809
HC at 12 months CGA, cm Boys Girls	46.5 (45.5–48.0) 45.7 (44.9–47.0)	46.5 (46.0–47.5) 45.3 (45.0–46.3)	0.526 0.739
**Nutrition**			
Duration of parenteral nutrition, days	7 (6–9)	5 (4–6)	<0.001
HM fortification started, days after birth	9 (7–11)	7 (6–8)	<0.001
Total protein, g/kg/day	3.1 (2.9–3.4)	3.2 (2.9–3.4)	0.747
Total carbohydrates, g/kg/day	13.3 (11.5–14.1)	13.0 (12.4–13.8)	0.863
Total fat, g/kg/day	5.1 (4.6–5.9)	5.0 (4.4–5.6)	0.641
Total energy, kcal/kg/day	115 (102–125)	114 (107–121)	0.839
**Neurodevelopment at 12 months CGA**			
MDI	94.6 (±11.4)	94.2 (±11.6)	0.991
PDI	86.2 (±13.6)	86.5 (±12.2)	0.868

Total nutrient and energy values are average daily intake, both parenteral and enteral, during the first 28 days of life. Values are median (interquartile range) and mean (standard deviation).

**Table 3 nutrients-14-03682-t003:** Distribution of MDI and PDI scores in groups. Values are number of infants (percentage). * MDI and PDI data were missing for one infant in EP group, MDI data were missing for two infants and PDI data for one infant in VP group.

Bayley II Scores	Mental Development Index		Psychomotor Development Index	
	Extremely Preterm (*n* = 40 *)	Very/Moderately Preterm (*n* = 52 *)	Extremely Preterm (*n* = 40 *)	Very/Moderately Preterm (*n* = 53 *)
Normal (≥85)	31 (75.6)	40 (74.1)	24 (58.5)	29 (53.7)
Mild (70–84)	8 (19.5)	11 (20.4)	14 (34.1)	20 (37.0)
Moderate/severe (<70)	2.4 (1)	1.9 (1)	2 (4.9)	4 (7.4)

**Table 4 nutrients-14-03682-t004:** Relationship between growth during hospitalization period, average daily total nutrient intake during first 28 days, and neurodevelopment at 12 months of CGA.

Model 1	MDI			PDI		
	R = 0.629, R^2^ = 0.395			R = 0.526, R^2^ = 0.277		
23–27 Weeks	B	β_s_	*p*	B	β_s_	*p*
Sex	−3.19	−0.14	0.407	−0.50	−0.02	0.921
Gestational age, weeks	−0.36	−0.04	0.847	2.10	0.17	0.386
Birth weight SDS	−2.97	−0.32	0.081	0.15	0.01	0.944
Change of weight SDS birth to discharge	−0.11	−0.01	0.963	−0.94	−0.08	0.757
Change of length SDS birth to discharge	1.11	0.14	0.488	1.99	0.20	0.341
Change of HC SDS birth to discharge	1.10	0.07	0.696	0.85	0.05	0.817
Total protein, g/kg/day	−3.23	−0.12	0.635	−2.48	−0.08	0.779
Total carbohydrates, g/kg/day	4.12	0.60	0.017	2.20	0.27	0.308
Total fat, g/kg/day	0.52	0.04	0.838	2.58	0.17	0.434
	**MDI**			**PDI**		
	**R = 0.376, R^2^ = 0.142**			**R = 0.396, R^2^ = 0.156**		
**28–34 Weeks**	**B**	**β_s_**	** *p* **	**B**	**β_s_**	** *p* **
Sex	4.92	0.21	0.207	−0.27	−0.01	0.946
Gestational age, weeks	−1.14	−0.16	0.619	−0.33	−0.05	0.887
Birth weight SDS	−2.35	−0.31	0.373	−0.88	−0.11	0.741
Change of weight SDS birth to discharge	−1.96	−0.13	0.607	−7.38	−0.47	0.063
Change of length SDS birth to discharge	1.88	0.17	0.437	4.16	0.38	0.096
Change of HC SDS birth to discharge	2.19	0.14	0.475	2.01	0.12	0.520
Total protein, g/kg/day	−3.76	−0.15	0.538	6.78	0.25	0.279
Total carbohydrates, g/kg/day	1.99	0.24	0.344	0.47	0.06	0.827
Total fat, g/kg/day	−1.19	−0.11	0.590	−1.24	−0.11	0.581

**Table 5 nutrients-14-03682-t005:** Relationship between growth after discharge, average daily total nutrient intake during first 28 days, and neurodevelopment at 12 months of CGA.

Model 2	MDI			PDI		
	R = 0.648, R^2^ = 0.420			R = 0.602, R^2^ = 0.362		
23–27 Weeks	B	β_s_	*p*	B	β_s_	*p*
Sex	−2.58	−0.11	0.476	1.12	0.04	0.804
Gestational age, weeks	0.88	0.09	0.644	3.32	0.27	0.167
Birth weight SDS	−2.08	−0.22	0.218	2.31	0.21	0.273
Change of weight SDS discharge to 12 months	−2.02	−0.29	0.300	−0.62	−0.08	0.797
Change of length SDS discharge to 12 months	1.67	0.18	0.420	3.11	0.28	0.231
Change of HC SDS discharge to 12 months	3.08	0.28	0.227	2.73	0.21	0.387
Total protein, g/kg/day	1.96	0.07	0.767	6.37	0.20	0.443
Total carbohydrates, g/kg/day	3.38	0.49	0.053	0.54	0.07	0.799
Total fat, g/kg/day	0.08	0.01	0.973	1.07	0.07	0.722
	**MDI**			**PDI**		
	**R = 0.431, R^2^ = 0.186**			**R = 0.376, R^2^ = 0.141**		
**28–34 Weeks**	**B**	**β_s_**	** *p* **	**B**	**β_s_**	** *p* **
Sex	6.39	0.28	0.096	2.33	0.10	0.565
Gestational age, weeks	−1.18	−0.17	0.566	1.13	0.16	0.605
Birth weight SDS	−2.97	−0.39	0.247	2.09	0.27	0.444
Change of weight SDS discharge to 12 months	−1.99	−0.22	0.331	−1.53	−0.16	0.483
Change of length SDS discharge to 12 months	−2.00	−0.22	0.39	−0.19	−0.02	0.939
Change of HC SDS discharge to 12 months	5.55	0.37	0.056	6.01	0.39	0.053
Total protein, g/kg/day	−7.02	−0.27	0.248	2.98	0.11	0.644
Total carbohydrates, g/kg/day	1.84	0.23	0.365	0.41	0.05	0.849
Total fat, g/kg/day	−1.77	−0.16	0.368	−2.97	−0.26	0.161

## Data Availability

The data generated and analyzed during the current study are not publicly available, but are available from the corresponding author on reasonable request.

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
