# Peer review of "Impact of Early Nutrient Intake and First Year Growth on Neurodevelopment of Very Low Birth Weight Newborns"

_nutrients, 2022, doi:10.3390/nu14183682_

Round 1
Reviewer 1 Report
The Authors performed a study to evaluate the relationship between early nutritional intake, first year growth and neurodevelopment.
To my opinion the manuscript has the possibility to be published in this prestigious Journal, however it needs to be improved. I have some comment to help the improvement of the manuscript
Title:
“Early Enteral Feeding – Impact of Nutrient Intake and First 2 Year Growth on Neurodevelopment of Very Low Birth Weight 3 Newborns”. In the article the nutritional intake used in the statistical analysis is the total amount, enteral and parenteral. I suggest to modify “Impact of Nutrient Intake and First 2 Year Growth on Neurodevelopment of Very Low Birth Weight 3 Newborns”.
Introduction:
- lines 39-41 need a reference;
- from line 57. You talk about early aminoacids intake. Recently it ha been demonstrated the different effects of enteral and parenteral nutrition on cerebral growth of preterm newborns, and the negative influence of early energy intake given by parenteral nutrition on the occurrence of metabolic side effects. This side effects are, in turn, associated with worse neurodevelopmental outcome and early enteral feeding reduce the risk of this metabolic side effects related to parenteral nutrition. I suggest to add this topic in introduction section and discuss it in discussion section.
Boscarino et al SciRep 2021 https://doi.org/10.1038/s41598-021-98088-4
Boscarino et al Nutrients 2021 https://doi.org/10.3390/nu13113886
Boscarino et al Nutrients 2021 https://doi.org/10.3390/nu13061930
Moltu et al JPGN 2021 https://doi.org/10.1097/mpg.0000000000003076
Terrin et al PlosOne 2020 https://doi.org/10.1371/journal.pone.0235540
Materials and methods:
- lines 83-88 and 102-103, figure 1 and table 1. I suggest to move them in results section;
- line 144 “number and percentage”
Results:
- It is not clear if there is difference or not between the growth parameters in each time points. In addition, in figure 1 “36w*” * what’s mean?
- Table 3 are not very clear. Maybe you should modify using two different columns for MDI and PDI
Discussion:
- I suggest to discuss also the topic of the influence of energy intakes (see comments above)
- lines 214-215, not for one years?
- You comment (lines 246-249) the possible influence of IUGR in neurodevelopmental outcome of preterm newborns. However, you did not evaluate this factor. Please add as limit
- lines 277-279. Not only breast milk, but in general enteral nutrition has a different, specifically positive, effects on brain growth.
Boscarino et al SciRep 2021 https://doi.org/10.1038/s41598-021-98088-4
-Limits section should be improved. There is not evaluated the different effects of parenteral and enteral nutrition, low sample size, there are other variables not evaluated that influence the neurological outcome and Bayley II has important limitations.
Picciolini et al BMCPed 2015 https://doi.org/10.1186/s12887-015-0457-x
Reviewer 2 Report
Thank you for allowing me to review this manuscript. This topic is timely and of interest to practicing clinicians within the neonatal field. My comments and suggestions are as follows:
§ Line 22: is 1 year follow-up from time of birth or at 1 year corrected age?
§ Line 27: “Growth before discharge was slower in EP group than VP.”—is this in grams/day or grams/kg/day?
§ Line 29: “Higher carbohydrate intake”—please provide a time period for this (e.g. the first week of life, first month of life, etc.)
§ Line 39-40: Please include a reference. Please also review the entire introduction carefully to see where there is opportunity to provide more references for statements made.
§ Line 70: Please describe what time period constitutes “early” nutritional intake
§ Line 72: The acronym “GA” is defined previously, but not “CGA”. Please review the entire manuscript and correctly include the appropriate term.
§ Line 72: Do you have any evidence to reference to support that infants from this unit receive “early progressive enteral feeding”? Otherwise, can you again provide your definition of “early”?
§ It is stated that parental consent was needed from both parents. How did you deal with single mothers?—e.g. where they not included?
§ Line 101: Is donor milk pasteurized or commercially sterilized?
§ Line 101: Please refer to reference 23 for “milk was advanced rapidly”.
§ Line 102: This information should be in the results section.
§ Line 113: What does “regular formula mean” (e.g. standard term infant formula at 20 calories/ounce?)?
§ I reviewed reference 23, which is your previously published article. I kindly ask that you include more details of parenteral nutrition management within this current unpublished manuscript (reference 23 provide sufficient detail for enteral feedings, but methods of parenteral management are insufficient to be replicated). For example, please include details on starting dosing for dextrose, protein, and intralipids with details on goals and how quickly these are advanced to goal.
§ Section 2.3: Can you please explain why you did not track infants on the 2013 Fenton growth chart or the WHO 0-2 year growth chart? There is much data on these and they are commonly used, so I’m interested in your perspective. You could evaluate birth to discharge z-scores and z-scores changes during NICU hospitalization from the 2013 Fenton growth curve, then may evaluate the same z-scores on the WHO 0-2 year for adjusted age from NICU discharge to 1 year corrected age.
§ Section 2.5: Please include what p-value was considered statistically significant.
§ I’m wondering why weight “SDS” was used instead of the traditional z-score? I would personally consider z-score instead as standard terminology, especially when evaluating changes in z-score with consideration of this manuscript: “Identifying Malnutrition in Preterm and Neonatal Populations: Recommended Indicators” by Goldberg et al., 2018.
§ Line 164: Please change “till” to “until”.
§ Line 164: “At 12 months”—please clarify if this is 12 months of age or 12 months corrected age.
§ Table 2 under “Nutrition”—please provide a reference for the estimated macronutrient intake. Is this only while on parenteral nutrition? Is this the average?
§ Please clarify, in the first one week of life, are estimated macronutrient or energy intakes in “per kg/day” based on birth weight (while infants are diuresing) or on actual (diuresed) daily weight?
§ Table 4: Readers have no context for the timing (and adjustment for) the macronutrients in per kg/day (e.g. first week of life, first month of life, during NICU hospitalization, etc). Likewise, we have no context if this is from enteral nutrition, parenteral nutrition, or a combination of the two. Again, please define the meaning of “early” intakes.
§ Line 217-220: Please include references for statements like “recommended nutrient intakes were reached”. Similarly, there was no inclusion in the Introduction or Methods section that indicate goal intakes.
§ Line 224: Please change “despite of” to “despite”.
§ Line 266: Can you provide objective rationale for why you believe this cohort “grew well”? Perhaps by z-score changes, etc.
§ Discussion section: One consideration surrounding carbohydrates is to consider differences in glucose levels. It is like the extremely preterm infants are more prone to hyperglycemia, etc. than very/moderately preterm infants. Thus another variable that may impact neurodevelopment.
§ It would be interesting to evaluate growth in grams/kg/day from birth to discharge.
Author Response
Please see the atachment.

Round 2
Reviewer 1 Report
Congratulations!